# Comparative analysis of Kernel-based versus BFGS-ANN and deep learning methods in monthly reference evapotranspiration estimation

Mohammad Taghi SATTARI [1,2,3], Halit APAYDIN[3], Shahab S. BAND[4], Amir MOSAVI[5,6,7], Ramendra PRASAD[8]

[1]Department of Water Engineering, Faculty of Agriculture, University of Tabriz, Tabriz 51666, Iran
[2]Institute of Research and Development, Duy Tan University, Danang 550000, Vietnam.
[3]Department of Agricultural Engineering, Faculty of Agriculture, Ankara University, Ankara 06110, Turkey
[4]Future Technology Research Center, National Yunlin University of Science and Technology, Douliou, Yunlin 64002, Taiwan
[5]Faculty of Civil Engineering, Technische Universität Dresden, 01069 Dresden, Germany
[6]John von Neumann Faculty of Informatics, Obuda University, 1034 Budapest, Hungary
[7]School of Economics and Business, Norwegian University of Life Sciences, 1430 Ås, Norway
[8]Department of Science, School of Science and Technology, The University of Fiji, Lautoka, Fiji

*Correspondence to*: Mohammad Taghi SATTARI (mtsattar@tabrizu.ac.ir and mohammadtaghisattari@duytan.edu.vn), Shahab S. Band (shamshirbands@yuntech.edu.tw)

**Abstract.** Timely and accurate estimation of reference evapotranspiration ($ET_0$) is indispensable for agricultural water management for efficient water use. This study aims to estimate the amount of $ET_0$ with machine learning approaches by using minimum meteorological parameters in the Corum region, which has an arid and semi-arid climate and is regarded as an important agricultural center of Turkey. In this context, monthly averages of meteorological variables *i.e.*, maximum and minimum temperature, sunshine duration, wind speed, average, maximum, and minimum relative humidity are used as inputs. Two different kernel-based methods i.e., Gaussian Process Regression (GPR) and Support Vector Regression (SVR)), together with BFGS-ANN, and Long short-term memory (LSTM) models were used to estimate $ET_0$ amounts in 10 different combinations. The results showed that all four methods predicted $ET_0$ amounts with acceptable accuracy and error levels. BFGS-ANN model showed higher success ($R^2 = 0.9781$) than the others. In kernel-based GPR and SVR methods, Pearson VII function-based universal kernel was the most successful ($R^2 = 0.9771$). Scenario 5 having temperatures including average temperature, maximum and minimum temperature, and sunshine duration as inputs gave the best results. The second-best scenario was with only the sunshine duration as the input to the BFGS-ANN which estimated $ET_0$ having a correlation coefficient of 0.971 (Scenario 8). Conclusively, this study shows the better efficacy of the BFGS in ANN for enhanced performance of the ANN model in $ET_0$ estimation for arid and semi-arid drought-prone regions.

## 1 Introduction

Accurate estimation of reference crop evapotranspiration ($ET_0$) and crop water consumption (ET) is essential in managing water in the agricultural sector particularly for arid and semi-arid climatic conditions where water is scarce and valuable. Although $ET_0$ is a complex element of the hydrological cycle, it is also an important component of hydro-ecological applications and water management in the agricultural sector. The estimation of $ET_0$ is critical in the forcible management of irrigation and hydro-meteorological studies on respective basins and on national scales (Pereira et al., 1999; Xu and Singh, 2001; Anli 2014) since knowledge of $ET_0$ would allow for reduced water wastage, increased irrigation efficiency, proper irrigation planning, and reuse of water.

In general, the equations that calculate $ET_0$ values are very complex, nonlinear, contain randomness, and all in all, have several underlying assumptions. The results obtained from these equations differ greatly from the measured values. $ET_0$ is considered a complex and nonlinear phenomenon that interacts with water, agriculture, and climate sciences. It is difficult to emulate such a phenomenon by experimental and classical mathematical methods. About twenty well-known methods for estimating $ET_0$ based on different meteorological variables and assumptions are available in the literature. The Penman-Monteith (FAO56PM) method proposed by FAO is recommended to estimate $ET_0$, as it usually gives usable results in different climatic conditions (Hargreaves and Samani, 2013; Rana and Katerji, 2000; Feng et al., 2016; Nema et al., 2017). Cobaner et al. (2016) modified the Hargreaves-Samani (HS) equation used in the determination of $ET_0$. Solving the equations and finding the correct parameter values requires sophisticated programs for the employment of differential equations, which require rigorous optimization methods together with a broad range of Spatio-temporal good quality and accurate input data with the knowledge of initial conditions (Prasad et al., 2017).

On the other hand, the developments in artificial intelligence (AI) methods and the increase in the accuracy of the estimation results have increased the desire for these AI methods. The AI models offer a number of advantages including; their ease of development compared to physically-based models; not requiring underlying boundary conditions or other assumptions or initial forcings; and has the ability to operate at localized positions (Prasad et al., 2020). Consequently, many studies have been reported to have applied AI approaches for $ET_0$ estimations. Artificial intelligence techniques based on machine learning (ML) has been successfully utilized in predicting complex and nonlinear processes in natural sciences, especially hydrology (Koch et al., 2019, Prasad and Deo, 2017; Solomatine, 2002; Solomatine and Dulal, 2003; Yaseen et al., 2016; Young et al. , 2017). Thus, methods such as ML and deep learning have gained popularity in estimating and predicting $ET_0$.

The artificial neural network (ANN) has been the widely used ML model to date. Sattari et al. (2013) used the backpropagation algorithm of ANN and tree-based M5 model to estimate the monthly $ET_0$ amount by employing a climate dataset (air temperature, total sunshine duration, relative humidity, precipitation, and wind speed) in the Ankara region and compared the estimated $ET_0$ with FAO56PM computations. The results revealed that the ANN approach gives better results. In another study, Pandey et al. (2017) in their study, ML techniques for $ET_0$ estimation using limited meteorological data; evaluated evolutionary regression (ER), ANN, multiple nonlinear regression (MLNR), and SVM and found the ANN FAO56PM model performing

better. In their study, Nema et al. (2017) studied the possibilities of using ANN to increase monthly evapotranspiration prediction performance in the humid area of Dehradun. They developed different ANN models, including combinations of various training functions and neuron numbers, and compared them with $ET_0$ calculated with FAO56PM. They found that the ANN trained by the Levenberg-Marquardt algorithm with 9 neurons in a single hidden layer made the best estimation performance in their case. The ANN, with multiple linear regression (MLR), ELM, and Hargreaves Samani models were tested by Reis et al. (2019) to predict $ET_0$ in the presence of temperature data in the Verde Grande River basin, Brazil. The study revealed that AI methods have superior performance over other models. Abrishami et al. (2019) estimated the amount of daily $ET_0$ for wheat and corn using ANN and found the proper and acceptable performance of ANNs with two hidden layers. However, some studies showed a slightly better performance of other models. Citakoglu et al. (2014) predicted monthly average $ET_0$ using the ANN and adaptive network-based fuzzy inference system (ANFIS) techniques using combinations of long-term average monthly climate data such as wind speed, air temperature, relative humidity, and solar radiation as inputs and found ANFIS to be slightly better than ANN. Yet recommended both methods to be successfully used in estimating the monthly mean $ET_0$. Likewise, ANN and ANFIS models by employing the Cuckoo search algorithm (CSA) were applied by Shamshirband et al. (2016) using data from twelve meteorological stations in Serbia. The results showed that the hybrid ANFIS-CSA could be employed for high-reliability $ET_0$ estimation.

Despite ANNs being universal approximators having the ability to approximate any linear or nonlinear system without being constrained to a specific form, it has some inherent disadvantages. Slow learning speed, over-fitting, and constrained in local minima with relatively tedious to determine key parameters, such as training algorithms, activation functions, and hidden neurons. These inherent structural problems sometimes make it difficult in adopting for applications. However, despite all the disadvantages, it is still a preferred method in all branches of science and especially in hydrology. Having said that, in this study, the ANN is benchmarked with other comparative models. One such model is support vector machine (SVM) developed by Vapnik (2013). SVMs have good generalization ability since it utilizes the concept of structural risk minimization hypothesis in minimizing both empirical risk and the confidence interval of the learning algorithm. Due to the underlying solid mathematical foundation of statistical learning theory giving it an advantage, the SVMs have been preferred in a number of studies and produced highly competitive performances in real-world applications (Quej et al., 2017). Subsequently, Wen et al. (2015) predicted daily $ET_0$ via SVM, using a limited climate dataset in the Ejina Basin, China using the highest and lowest air temperatures, daily solar radiation and wind speed values as model inputs and FAO56PM results as model output. The SVM method's performance was compared to ANN and empirical techniques, including Hargreaves, Priestley-Taylor, and Ritchie, which revealed that the SVM recorded better performance. Zhang et al. (2019) examined SVM's success in $ET_0$ estimation and compared the outcomes with Hargreaves, FAO-24, Priestley-Taylor, McCloud, and Makkink. SVM was determined to be the most successful model. However, SVM also has several drawbacks, such as high computational memory requirement as well as being computational exhaustive as a large amount of computing time during the learning process is necessary.

In order to overcome the disadvantages of these two widely accepted approaches (ANN and SVM), many new modelling techniques have been proposed in recent years. For instance, the two state-of-the-art machine learning techniques, namely

Gauss Process Regression (GPR) and long short-term memory (LSTM) are also being recently trialled in the hydrologic time series modelling and forecasting applications. Following the newer developments, Shabani et al. (2020) used ML methods, including GPR, random forest (RF), and SVR, with meteorological inputs to estimate evaporation (PE) in Iran and found that ML methods have high performances even with a small number of meteorological parameters. In a recent study, deep learning and ML techniques to determine daily $ET_0$ have been explored in Punjab's Hoshiarpur and Patiala regions, India (Saggi et al., 2019). They found that supervised learning algorithms such as the deep learning-multilayer sensors (DL) model offers high performance for daily ET0 modelling. However, to the best of the author's knowledge, there have been very few attempts to test the practicability and ability of these two advanced approaches (LSTM and GPR) for $ET_0$ modelling and prediction. In addition, many studies included solar radiation in the modelling process, yet did not include sunshine hours in the modelling, which will be dealt with in this study.

With recent developments in ML methods with the use of deep learning techniques such as LSTM in water engineering together with technical developments in computers and the emergence of relatively comfortable coding languages, this study explores the application of different deep learning (LSTM) and other machine learning methods (ANN, SVM and GPR) in the estimation of $ET_0$ to shed light on future research and to determine effective modelling approaches relevant to this field. $ET_0$ is one of the essential elements in water, agriculture, hydrology, and meteorology studies, and its accurate estimation has been an open area of research due to $ET_0$ being a complex and nonlinear phenomenon. Hence, robust deep learning and ML approaches including LSTM, ANN, SVM and GPR methods need to be aptly tested. As a result, this study has three important goals; i) to estimate the amount of $ET_0$ using deep learning and machine learning methods, i.e., GPR, SVR, ANN employing Broyden–Fletcher–Goldfarb–Shanno (BFGS-ANN) learning algorithm, and LSTM in Corum conditions with a total annual rainfall of 427 mm classed as an arid and semi-arid climatic region; ii) to investigate the effect of different kernel functions of the SVR and GPR models on the performance of $ET_0$ estimation and; iii) to determine the model that provides the highest performance with the least meteorological variable requirement for the study. A proper prediction of reference evapotranspiration would be vital in managing limited water resources for optimum agricultural production.

## 2 Study area and dataset used

Corum's encompasses an area of 1 278 381 ha, of which 553 011 ha, or 43%, is agricultural land (Figure 1). Its population is 525 180 and 27% of it lives in rural areas. The city's water resource potential is 4 916 $hm^3$/year and 84 988 ha of agricultural land is being irrigated. The main agricultural products are wheat, paddy, chickpeas, onions, walnuts, and green lentils. This study was conducted using monthly meteorological data including highest and lowest temperature, sunshine duration, wind speed, average, highest, and lowest relative humidity from January 1993 - December 2018 (Anonymous, 2017) as model inputs leading to 312 months. 200 months were used for training, and the remaining 112 were used for testing. Statistics of the data used are given in Table 1. During the training period, the daily average, highest, and lowest temperature averages are 10.80, 18.27, and $4.02^0C$, respectively. The average sunshine duration in the region is 6.29 hours, wind speed is 1.72 m/s, and the

mean humidity is 70.41%. The lowest skewness coefficient was found in RHmax with -0.64 and the highest in RHmin

parameter with 0.35. The lowest kurtosis coefficient has Tmean with -1.24 and the highest with 1.12 by RHmax parameter.

The highest variation was observed in RHmin with 140.40 and the lowest in sunshine duration with 0.18. Similarly in the test

period, the daily average, highest, and lowest temperature averages are $11.44^0$C, $18.60^0$C, and $4.89^0$C, respectively. The

average sunshine duration in the region is 5.74 hours, wind speed is 1.64 m/s, and the mean humidity is 68.08%. The lowest

skewness coefficient was found in RHmax with -0.53 and the highest in RHmin parameter with 0.75. The lowest kurtosis

coefficient has Tmean with -1.25 and the highest with -0.37 by RHmax and RHmin parameters. The highest variation was

observed in RHmin with 202.50 and the lowest in sunshine duration with 0.16. The skewness and kurtosis coefficients in the

train and the test period are similar in all parameters except the maximum relative humidity. The frequency distributions of

meteorological data of the study area are given in Figure 2 which conforms to the distribution statistics. As it is understood

from the figure, the dependent variable $ET_0$ values do not conform to the normal distribution.

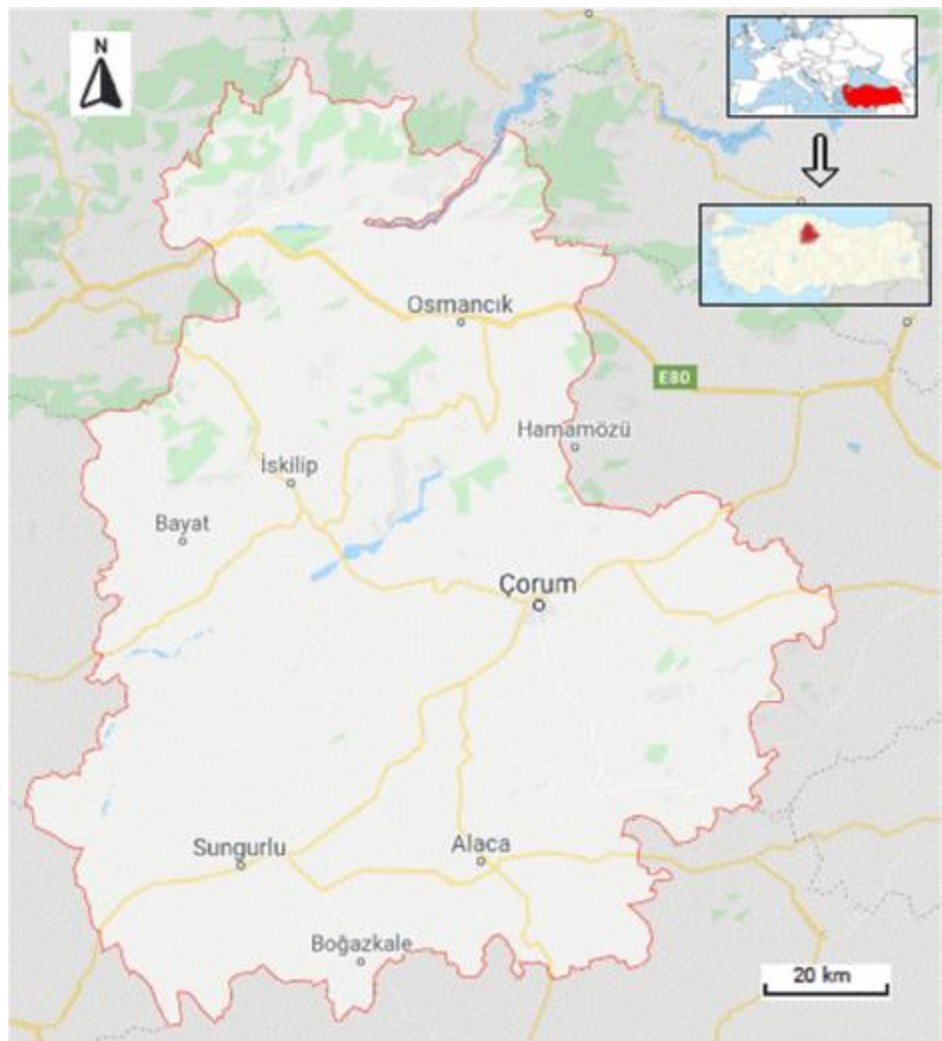

Figure 1. Location of the study area, Corum Province, Turkey (© Google Maps.)

Table 1. Basic statistics of the data used in the study during the training and testing periods

| Period | Statistic | Tmean ($^0$C) | Tmax ($^0$C) | Tmin ($^0$C) | $n$ (hr) | $U$ (m/s) | RHmean (%) | RHmax (%) | RHmin (%) | ET$_0$ (mm/month) |
|---|---|---|---|---|---|---|---|---|---|---|
| Training data set | Minimum | -6.18 | -1.27 | -11.3 | 1 | 0.95 | 51.6 | 66.87 | 21.51 | 11.76 |
| | Maximum | 25.06 | 35.44 | 14.75 | 11.97 | 2.69 | 94.74 | 98.93 | 82.83 | 185.59 |
| | Mean | 10.80 | 18.27 | 4.02 | 6.29 | 1.72 | 70.41 | 87.76 | 47.48 | 79.15 |
| | Stdev | 8.00 | 9.32 | 6.34 | 2.96 | 0.42 | 8.02 | 5.39 | 11.88 | 52.64 |
| | Skewness | -0.09 | -0.15 | -0.13 | 0.06 | 0.13 | 0.16 | -0.64 | 0.35 | 0.34 |
| | Kurtosis | -1.24 | -1.21 | -1.06 | -1.25 | -0.85 | -0.37 | 1.12 | -0.48 | -1.29 |
| | Coefficient of variation | 63.75 | 86.50 | 40.02 | 8.72 | 0.18 | 63.97 | 28.86 | 140.40 | 2756.72 |
| | Number of records | 200 | 200 | 200 | 200 | 200 | 200 | 200 | 200 | 200 |
| Testing data set | Minimum | -4.25 | 1.08 | -9.21 | 0.83 | 0.7 | 45.8 | 72.06 | 19.03 | 13.99 |
| | Maximum | 25.06 | 34.85 | 15.63 | 10.87 | 2.45 | 94.07 | 99.83 | 80.12 | 180.53 |
| | Mean | 11.44 | 18.60 | 4.89 | 5.74 | 1.64 | 68.08 | 90.09 | 40.53 | 79.21 |
| | Stdev | 7.82 | 9.17 | 6.23 | 2.92 | 0.39 | 11.23 | 6.21 | 14.17 | 53.02 |
| | Skewness | -0.04 | -0.15 | -0.03 | 0.08 | 0.08 | 0.25 | -0.53 | 0.75 | 0.36 |
| | Kurtosis | -1.25 | -1.20 | -1.12 | -1.23 | -0.65 | -0.74 | -0.37 | -0.37 | -1.27 |
| | Coefficient of variation | 61.68 | 84.89 | 39.20 | 8.60 | 0.16 | 127.17 | 38.90 | 202.50 | 2836.65 |
| | Number of records | 112 | 112 | 112 | 112 | 112 | 112 | 112 | 112 | 112 |

NB: T: Temperature, $n$: Sunshine duration, $U$: Wind speed, RH: Relative humidity

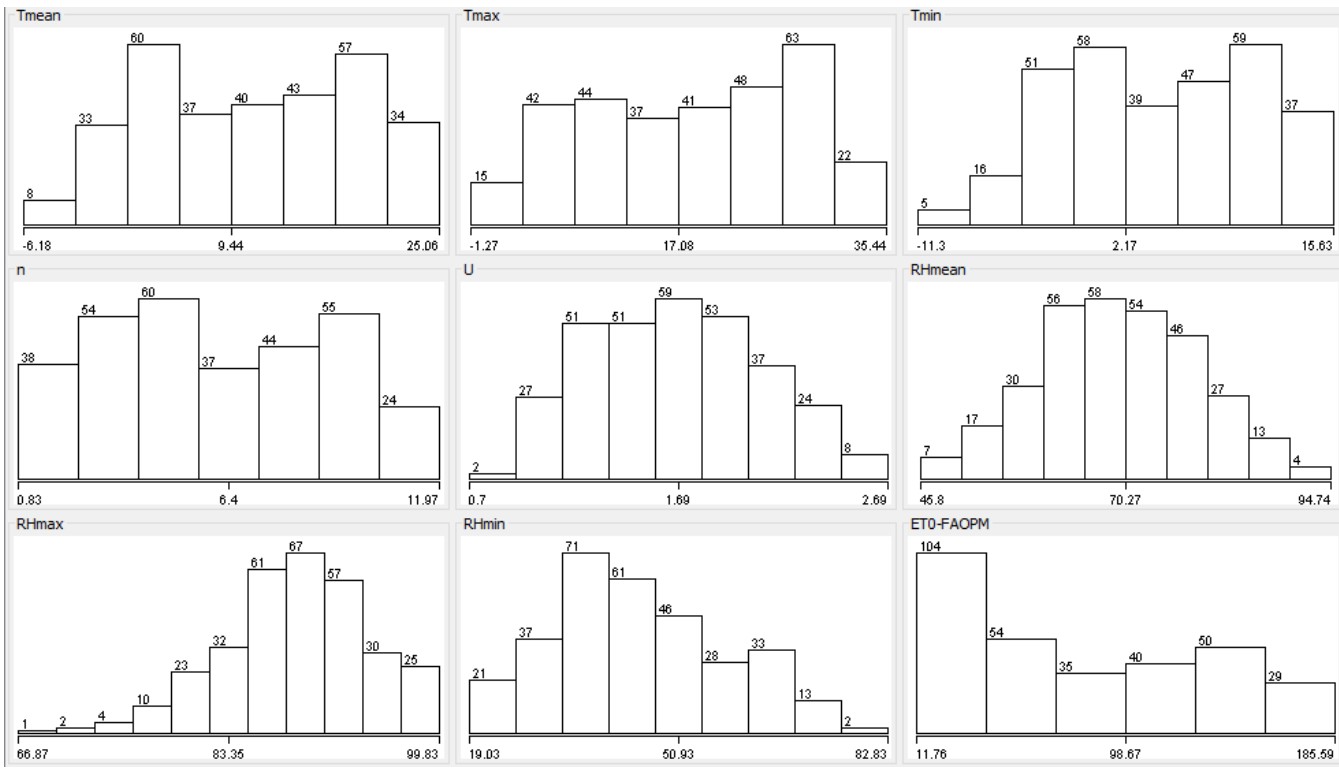

Figure 2. Frequency distributions of meteorological input data set conforming to the distribution statistics.

To determine the meteorological factors employed in the model and the formation of scenarios, the relationship between $ET_0$ and other variables were calculated as given in Figure 3. Input determination is an essential component of model development as irrelevant inputs are likely to worsen the model performances (Hejazi and Cai, 2009; Maier and Dandy, 2000; Maier et al., 2010), while a set of carefully selected inputs could ease the model training process and increase the physical representation

whilst providing a better understanding of the system (Bowden et al., 2005). The Sunshine duration in this study was very highly correlated with $ET_0$ ($R^2 = 0.92$) together with the variables Tmean, Tmax and Tmin were all highly correlated ($R^2 > 0.8$). The RH mean was the least correlated variable ($R^2 = 0.24$) in this study. As can be understood visually, the meteorological variables associated with temperature and especially the sunshine duration has a high correlation with $ET_0$. Considering these relationships, ten different input scenarios were created, and the effect of meteorological variables on $ET_0$ estimation was

evaluated. Table 2 gives the meteorological variables used in each scenario. While all parameters were taken into account in the first scenario, the ones that could affect $ET_0$ more in the following scenarios were added in the respective scenarios.

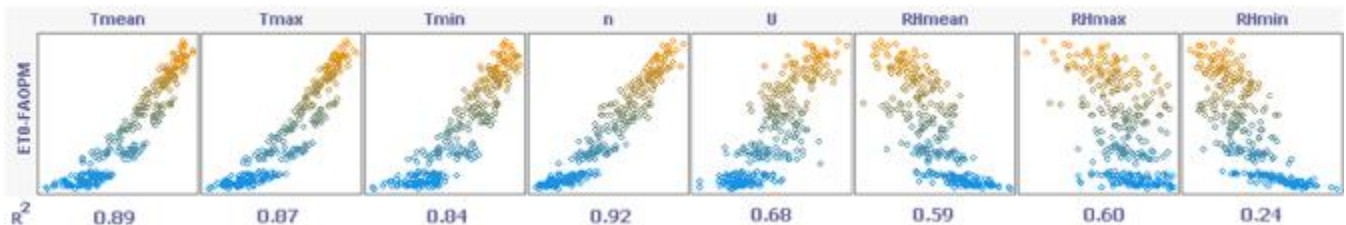

Figure 3. Scatter plot showing the correlation between $ET_0$ and the independent variable. The coefficient of determination has been added for clarity.

Table 2. Illustrates the scenarios developed in this study with respective inputs in respective scenarios.

| Scenario | Inputs |
|---|---|
| 1 (All Variables) | TMean, TMax, TMin, $n$, $U$, RHMax, RHMin, RHMean |
| 2 | TMean, $n$, $U$, RHMean |
| 3 | TMax, $n$, RHMax |
| 4 | TMax, $n$ , $U$ |
| 5 | TMean, TMax, TMin, $n$ |
| 6 | $n$, $U$, RHMax |
| 7 | $n$, RHMax |
| 8 (Highest $R^2$) | $n$ |
| 9 | TMin |
| 10 | TMax |

## 3 Methods

### 3.1 Calculation of $ET_0$

The United Nations, Food and Agriculture Organization (FAO) recommend Penman-Monteith (PM) equation (Eq.1) to calculate the evapotranspiration of reference crops (Doorenbos and Pruitt, 1977). Although the PM equation is much more complex than the other equations, it has been formally explained by FAO. The equation has two main features: (1) It can be used in any weather conditions without local calibration, and (2) the performance of the equation is based on the lysimetric data in an approved spherical range (Allen et al., 1989). The requirement for many meteorological factors can be defined as the main problem. However, there is still no equipment to record these parameters correctly in many countries, or data is not regularly recorded (Gavili et al., 2018).

$$ET_0 = \frac{0.408\, \Delta\, (R_n - G) + \gamma \frac{900}{T+273} u_2 (e_s - e_a)}{\Delta + \gamma (1 + 0.34 u_2)} \qquad \text{(Eq.1)}$$

Where

ET$_0$ refers to the reference evapotranspiration [mm day$^{-1}$],

G refers to the soil heat flux density [MJ m$^{-2}$ day$^{-1}$],

u$_2$ refers to the wind speed at 2 m [m s$^{-1}$],

e$_a$ refers to the actual vapour pressure [kPa],

e$_s$ refers to the saturation vapour pressure [kPa],

e$_s$-e$_a$ refers to the saturation vapour pressure deficit [kPa],

T refers to the mean daily air temperature at 2 m [°C],

Rn refers to the net radiation at the crop surface [MJ m$^{-2}$ day$^{-1}$],

$\gamma$ refers to the psychrometric constant [kPa °C$^{-1}$],

$\Delta$ refers to the slope vapour pressure curve [kPa °C$^{-1}$].

## 3.2 Broyden– Fletcher – Goldfarb – Shanno Artificial Neural Networks (BFGS-ANN)

McCulloch and Pitts (1943) pioneered the original idea of neural networks. ANN is essentially a black-box modelling approach that does not identify the training algorithm explicitly, yet the modellers often trial several algorithms to attain an optimal model (Deo and Şahin, 2015). In this study, the Broyden – Fletcher – Goldfarb – Shanno (BFGS) training algorithm has been used to estimate ET$_0$ amounts. In optimization studies, the BFGS method is a repetitious approach for solving unlimited nonlinear optimization problems (Fletcher, 1987). The BFGS-ANN technique trains a multilayer perceptron ANN with one hidden layer by reducing the given cost function plus a quadratic penalty using the BFGS technique. The BFGS approach includes Quasi-Newton methods. For such problems, the required condition for reaching an optimal level occurs when the gradient is zero. Newton and the BFGS methods cannot be guaranteed to converge unless the function has a quadratic Taylor expansion near an optimum. However, BFGS can have a high accuracy even for non-smooth optimization instances (Curtis et al., 2015).

Quasi-Newton methods do not compute the Hessian matrix of second derivatives. Instead, the Hessian matrix is drawn by updates specified by gradient evaluations. Quasi-Newton methods are extensions of the secant method to reach the basis of the first derivative for multi-dimensional problems. The secant equation does not specify a specific solution in multi-dimensional problems, and Quasi-Newton methods differ in limiting the solution. The BFGS method is one of the frequently used members of this class (Nocedal and Wright, 2006). In the BFGS-ANN method application, all attributes, including the target attribute (meteorological variables and ET$_0$) are standardized. In the output layer, the sigmoid function is employed for classification.

In approximation, the sigmoidal function can be specified for both hidden and output layers. For regression, the activation function can be employed as the identity function in the output layer. This method was implemented on the basis of radial basis function networks trained in a fully supervised manner using WEKA's Optimization class by minimizing squared error with the BFGS method. In this method, all attributes are normalized into the [0,1] scale (Frank, 2014).

### 3.3 Support Vector Machine (SVR)

The statistical learning theory is the basis of the SVM. The optimum hyperplane theory and kernel functions and nonlinear classifiers were added as linear classifiers (Vapnik, 2013). Models of the SVM are separated into two main categories: (a) The classifier SVM and (b) the regression (SVR) model. An SVM is employed to classify data in various classes, and the SVR is employed for estimation problems. Regression is used to take a hyperplane suitable for the data used. The distance to any point in this hyperplane shows the error of that point. The best technique proposed for linear regression is the least-squares (LS) method. However, it may be entirely impossible to use the LS estimator in the presence of outliers. In this case, a robust predictor has to be developed that will not be sensitive to minor changes, as the processor will perform poorly. Three kernel functions were used including Polynomial, Pearson VII function-based universal, and radial basis function with the level of Gaussian Noise Parameters added to the diagonal of the covariance matrix and the random number of seed to be used (equal to 1.0); the most suitable kernel function in each scenario was determined by trial and error (Frank, 2014) and the description is provided in Section 3.6.

### 3.4 Gauss process regression (GPR)

The GPR or GP is defined by Rasmussen and Williams (2005) as a complex set of random variables, which have a joint Gaussian distribution. Kernel-based methods such as SVM and GPs can work together to solve flexible and applicable problems. The GP is generally explained by two functions: Average and covariance functions (Eq. 2). The average function is a vector; the covariance function is a matrix. The GP model is possibly a nonparametric black box technique.

$$f \approx GP\ (m,\ k) \qquad\qquad\qquad (Eq.\ 2)$$

Where $f$ refers to Gauss distribution, $m$ refers to a mean function and $k$ refers to covariance function.

The value of covariance expresses the correlation between the individual outputs concerning the inputs. The covariance value determines the correlation between individual outputs and inputs. The covariance function produces a matrix of two parts (Eq.3).

$$Cov\ (x_p) = C_f\ (x_p) + C_n\ (x_p) \qquad\qquad\qquad (Eq.\ 3)$$

Here, $C_f$ represents the functional part, but defines the unknown part of the modelling system, while $C_n$ represents the system's noise part. A Gaussian process (GP) is closely related to SVM, and both are part of the kernel machine area in ML models. Kernel methods are sample-based learners. Instead of learning a fixed parameter, the kernels memorize the training data sample and assign a certain weight to it.

## 3.5 Long short-term memory (LSTM)

LSTM is a high-quality evolution of Recurrent Neural Networks (RNN). This neural network is presented to address the problems that existed in RNN and are done by adding more interactions per cell. These systems are also special since it remembers information for an extended period. Moreover, it also includes four essential interacting layers, and all of them include different communication methods.

The next thing is that its complete network consists of a memory block. These blocks are also called cells. The information is stored in one cell and then transferred into the next one with the help of gate controls. Through the help of these gates, it becomes straightforward to analyze the information accurately. All of these gates are extremely important, and they are called forget gates as explained in Eq. 4.

$$f_t = \sigma\left(W_f[h_{t-1}, X_t] + b_f\right) \tag{Eq. 4}$$

LSTM units or blocks are part of the repetitive neural network structure. Repetitive neural networks are made to use some artificial memory processes that can help these AI algorithms to mimic human thinking.

## 3.6 Kernel functions

Four different kernel functions are frequently used as depicted in literature including the polynomial, radial-based function, Pearson VII function (PUK), and normalized polynomial kernels used and their formulas and parameters are tabulated in Table 3. As is clear from Table 3, some parameters must be determined by the user for each kernel function. While the number of parameters to be determined for PUK kernel is two, it requires determining a parameter in the model formation that will be the basis for classification for other functions. When kernel functions are compared, it is seen that polynomial and radial based kernels are more plain and understandable. Although it may seem mathematically simple, the increase in the degree of the polynomial makes the algorithm complex. This significantly increases processing time and decreases the classification accuracy after a point. In contrast, changes in the radial-based function parameter ($\gamma$), expressed as the kernel size were less effective on classification performance (Hsu et al., 2010). The normalized polynomial function was proposed by Arnulf et al. (2001) in order to normalize the mathematical expression of the polynomial kernel instead of normalizing the data set. The normalized polynomial kernel is a generalized version of the polynomial kernel. On the other hand, the PUK kernel has a more complex mathematical structure than other kernel functions with its two parameters ($\sigma$, $\omega$) known as Pearson width.

These two parameters affect classification accuracy and these parameters are not known in advance. For this reason, determining the most suitable parameter pair in the use of the PUK kernel is an important step.

Table 3. Basic kernel functions used in the study with parameters that needed to be determined.

| Kernel functions | Mathematical Expression | Parameter |
|---|---|---|
| **Polynomial kernel** | $K(x,y) = ((x.y) + 1)^d$ | Polynomial degree (d) |
| **Radial Based Function Kernel** | $K(x,y) = e^{-\gamma|(x-x_i)|^2}$ | Kernel size ($\gamma$) |
| **PUK** | $K(x,y) = \dfrac{1}{\left[1 + \left(\dfrac{2.\sqrt{\|x-y\|^2}\ \sqrt{2^{(1/\omega)}-1}}{\sigma}\right)^2\right]^{\omega}}$ | Pearson width parameters ($\sigma$, $\omega$) |

The user must determine the editing parameter C for all SVM during runtime. If values that are too small or too large for this parameter are selected, the optimum hyperplane cannot be determined correctly. Therefore there will be a serious decrease in classification accuracy. On the other hand, if C equal to infinity, the SVM model becomes suitable only for datasets that can

be separated linearly. As can be seen from here, the selection of appropriate values for the parameters directly affects the accuracy of the SVM classifier. Although a trial and error strategy is generally used, the cross-validation approach enables successful results. The purpose of the cross-validation approach is to determine the performance of the classification model created. For this purpose, the data is separated into two categories where the first is used as training the model and, the second part is processed as test data to determine the model's performance. As a result of applying the model created with the training

set to the test data set, the number of samples classified correctly indicates the classifier's performance. Therefore, by using the cross-validation method, the classification and determination of the best kernel parameters were obtained (Kavzoglu and Golkesen, 2010).

In this study, during SVR and GPR modelling, the three kernel functions as in Table 3 were used and the most suitable kernel function in each scenario was determined by trial and error (Frank, 2014). For the BFGS-ANN, SVR, and GPR methods in the

315 Weka software were used, while python language was used for the LSTM method.

### 3.7 Model Evaluation

The statistical parameters used in the selection and comparison of the models in the study included the root mean square error (RMSE), mean absolute error (MAE), and correlation fit ($R$) as shown in Eq. 5-7. Here, $X_i$ and $Y_i$ are the observed and predicted values, and $N$ is the number of data.

$$MAE = \frac{1}{N}\Sigma_{i=1}^{N}|X_i - Y_i| \tag{Eq. 5.}$$

$$RMSE = \sqrt{\frac{1}{N}\Sigma_{i=1}^{N}(X_i - Y_i)^2} \tag{Eq. 6.}$$

$$R = \frac{N\sum X_i Y_i - (\sum X_i)(\sum Y_i)}{\sqrt{N(\sum X_i^2) - (\sum X_i)^2}\sqrt{N(\sum Y_i^2) - (\sum Y_i)^2}} \tag{Eq. 7.}$$

In addition, Taylor diagrams were prepared to check the performance of the models, which illustrates the experimental and statistical parameters simultaneously.

### 4 Results

In this study, 10 different scenarios were created by using combinations of input variables, i.e., monthly average, highest and lowest temperature, sunshine duration, wind speed, average, highest, and lowest relative humidity data. $ET_0$ amounts were estimated with the help of kernel-based GPR and SVR methods, BFGS-ANN, and one of the deep learning methods LSTM models. $ET_0$ estimation results obtained from different scenarios according to the GPR method are summarized in Table 4. As can be seen from the table, the 5th scenario containing four meteorological variables including TMax, TMin, TMean and $n$ with the GPR method PUK function gave the best result (Train period: $R^2$ = 0.9667, MAE = 9.1279 mm/month, RMSE = 11.067 mm/month; Test period: $R^2$ = 0.9643, MAE = 9.1947 mm/month, RMSE = 11.2109 mm/month). However, the 8th scenario with only one meteorological variable (sunshine duration) registered quite well results with training period: $R^2$ = 0.9472, MAE = 10.1629 mm/month, RMSE = 13.2694 mm/month and testing period: $R^2$ = 0.9392, MAE = 11.8473 mm/month, RMSE = 15.8719 mm/month. Since the scenario with the least input parameters and with an acceptable level of accuracy is largely preferred, scenario 8 was chosen as the optimum scenario.

The scatter plot and time series plots of the test phase for scenario 5 and 8 are given in Figures 4 and 5. As can be seen from these Figures, a relative agreement has been achieved between the FAO56PM $ET_0$ values and the $ET_0$ values modelled. When the time series graphs are examined, minimum points in estimated $ET_0$ values are more in harmony with FAO56PM values than maximum points.

Table 4. Outcomes of the GPR modelling approach from different kernel functions based on $R^2$, MAE, and RMSE (*Italics* represents the best results; Bold represents the optimally selected model).

| Scenario No | Kernel functions | Train | | | Test | | |
|---|---|---|---|---|---|---|---|
| | | $R^2$ | MAE | RMSE | $R^2$ | MAE (mm/month) | RMSE (mm/month) |
| 1 | Polynomial | 0.9084 | 13.1238 | 16.0365 | 0.8451 | 17.8013 | 21.4952 |
| | PUK | 0.9732 | 6.8024 | 8.9055 | 0.9506 | 10.5906 | 13.4330 |
| | Radial basis function | 0.9357 | 22.3706 | 25.3578 | 0.9220 | 22.3353 | 25.4332 |
| 2 | Polynomial | 0.8825 | 15.0049 | 18.3607 | 0.8332 | 19.4655 | 23.9183 |
| | PUK | 0.9666 | 7.2041 | 9.4750 | 0.9639 | *8.9058* | 11.5185 |
| | Radial basis function | 0.9450 | 27.7700 | 31.2897 | 0.9366 | 27.5940 | 31.2150 |
| 3 | Polynomial | 0.8697 | 15.7587 | 19.2936 | 0.7807 | 21.2623 | 26.2083 |
| | PUK | 0.9436 | 9.5556 | 12.6058 | 0.9335 | 12.2152 | 15.0187 |
| | Radial basis function | 0.9251 | 31.3045 | 35.4426 | 0.9073 | 31.9344 | 36.1935 |
| 4 | Polynomial | 0.7002 | 37.824 | 43.417 | 0.7105 | 36.6604 | 41.2745 |
| | PUK | 0.9637 | 7.7384 | 10.153 | 0.9629 | 9.3003 | 12.4647 |
| | Radial basis function | 0.9374 | 29.1996 | 32.9582 | 0.9491 | 29.7864 | 33.6709 |
| 5 | Polynomial | 0.6312 | 35.1424 | 40.3818 | 0.6030 | 33.8278 | 38.3742 |
| | *PUK* | *0.9667* | *9.1279* | *11.067* | *0.9643* | *9.1947* | *11.2109* |
| | Radial basis function | 0.9239 | 25.6568 | 29.4976 | 0.9239 | 26.2766 | 30.0768 |
| 6 | Polynomial | 0.8703 | 15.6789 | 19.3039 | 0.7841 | 21.5210 | 27.2959 |
| | PUK | 0.9569 | 8.5950 | 11.1225 | 0.9401 | 12.1685 | 15.8165 |
| | Radial basis function | 0.9229 | 33.0011 | 36.9189 | 0.8991 | 33.4845 | 37.9140 |
| 7 | Polynomial | 0.8599 | 16.6129 | 20.0640 | 0.7852 | 21.7258 | 26.9480 |
| | PUK | 0.9349 | 10.3820 | 13.5482 | 0.9310 | 12.9590 | 16.5650 |
| | Radial basis function | 0.9086 | 36.4501 | 40.9667 | 0.8746 | 36.9353 | 41.6716 |
| 8 | Polynomial | 0.9203 | 41.2839 | 46.5019 | 0.9281 | 40.4306 | 45.9593 |
| | **PUK** | **0.9472** | **10.1629** | **13.2694** | **0.9392** | **11.8473** | **15.8719** |
| | Radial basis function | 0.9283 | 37.0877 | 41.8535 | 0.9281 | 37.6298 | 42.3803 |
| 9 | Polynomial | 0.8394 | 44.0191 | 49.2989 | 0.8380 | 43.9357 | 50.0790 |
| | PUK | 0.8759 | 15.0361 | 18.5984 | 0.8634 | 16.2747 | 20.1854 |
| | Radial basis function | 0.8398 | 39.0547 | 44.3349 | 0.8380 | 40.0566 | 44.8850 |
| 10 | Polynomial | 0.8677 | 43.1716 | 48.2151 | 0.8746 | 42.6604 | 48.7584 |
| | PUK | 0.9027 | 13.3821 | 16.4932 | 0.9130 | 13.0145 | 15.8309 |
| | Radial basis function | 0.8679 | 38.2998 | 43.4373 | 0.8748 | 39.1677 | 43.9253 |

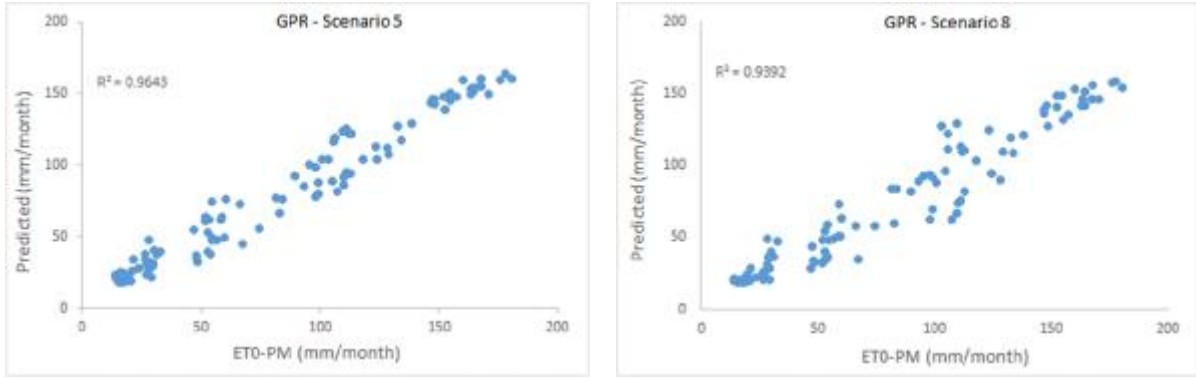

Figure 4. Scatter plots comparing GPR estimated and FAO56PM estimated $ET_0$ in scenarios 5 and 8

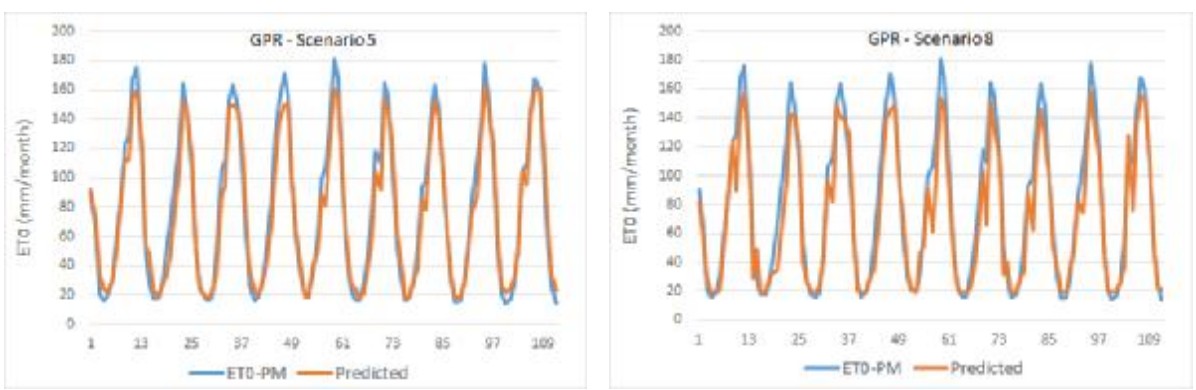

Figure 5. Time series graphics of GPR estimated and FAO56PM estimated $ET_0$ in scenarios 5 and 8

For the SVR model, again 3 different kernel functions were evaluated in respective scenarios under the same conditions, and the results are displayed in Table 5. As can be seen here, scenarios 5 and 8 have yielded the best and most appropriate results according to the PUK function. The results of the 5th scenario with TMean, TMin, TMax and $n$ as input variables gave the best result (Train period: $R^2 = 0.9838$, MAE = 6.0500 mm/month, RMSE = 8.5733 mm/month; Test period: $R^2 = 0.9771$, MAE = 7.07 mm/month, RMSE = 9.3259 mm/month). However, scenario 8 gave the most appropriate result (Train period: $R^2 = 0.9398$, MAE = 9.7984 mm/month, RMSE = 13.0830 mm/month; Test period: $R^2 = 0.9392$, MAE = 11.2408 mm/month, RMSE = 15.5611 mm/month) only with one meteorological input variable, i.e., the sunshine duration ($n$). Although the accuracy rate of the 8th scenario is somewhat lower than the 5th scenario, it provides convenience and is preferred in terms of application and calculation since it requires a single input. The sunshine duration can be measured easily and without the need for high-cost equipment and personnel. Consequently, by using only one parameter, the amount of $ET_0$ is estimated within acceptable accuracy limits.

Table 5. Outcomes of the SVR modelling approach from different kernel functions based on $R^2$, MAE, and RMSE (*Italics* represents the best results; Bold represents the optimally selected model).

| Scenario No | Kernel function | Train | | | Test | | |
|---|---|---|---|---|---|---|---|
| | | $R^2$ | MAE | RMSE | $R^2$ | MAE | RMSE |
| 1 | Polynomial | 0.9667 | 7.6671 | 9.6167 | 0.9655 | 11.0033 | 13.5740 |
| | PUK | 0.9790 | 1.3130 | 2.9310 | 0.9683 | 8.70480 | 11.1693 |
| | Radial basis function | 0.9446 | 10.3256 | 12.5561 | 0.9366 | 11.1203 | 13.4468 |
| 2 | Polynomial | 0.9587 | 9.8445 | 12.0674 | 0.9526 | 10.1138 | 11.6124 |
| | PUK | 0.9775 | 4.3655 | 8.0208 | 0.9742 | 8.88250 | 11.6469 |
| | Radial basis function | 0.9487 | 11.0557 | 12.8207 | 0.9456 | 11.4313 | 13.5386 |
| 3 | Polynomial | 0.9392 | 10.088 | 13.468 | 0.9160 | 13.5919 | 15.903 |
| | PUK | 0.9608 | 7.1018 | 7.1018 | 0.9249 | 12.0206 | 15.6733 |
| | Radial basis function | 0.9401 | 12.1973 | 14.4483 | 0.9107 | 15.1051 | 18.4364 |
| 4 | Polynomial | 0.9491 | 10.5076 | 12.7585 | 0.9485 | 11.8516 | 14.1386 |
| | PUK | 0.9732 | 5.5868 | 8.6784 | 0.9604 | 9.2452 | 12.5707 |
| | Radial basis function | 0.9593 | 12.7177 | 14.8832 | 0.9500 | 12.6226 | 16.1700 |
| 5 | Polynomial | 0.9743 | 8.9452 | 11.5497 | 0.9657 | 8.5349 | 10.2108 |
| | *PUK* | *0.9838* | *6.0500* | *8.5733* | *0.9771* | *7.0700* | *9.3259* |
| | Radial basis function | 0.9414 | 11.8017 | 15.1588 | 0.9318 | 11.8607 | 14.4412 |
| 6 | Polynomial | 0.9399 | 10.3413 | 12.9082 | 0.9281 | 14.5901 | 17.9626 |
| | PUK | 0.9698 | 6.1970 | 9.1435 | 0.9497 | 11.2859 | 14.7455 |
| | Radial basis function | 0.9299 | 13.9103 | 17.0013 | 0.9120 | 16.7198 | 22.2031 |
| 7 | Polynomial | 0.9214 | 11.9563 | 14.8277 | 0.9214 | 14.7185 | 17.6297 |
| | PUK | 0.9426 | 9.1560 | 12.6111 | 0.9407 | 12.0180 | 15.5924 |
| | Radial basis function | 0.9164 | 17.8134 | 21.4555 | 0.8951 | 19.4352 | 25.7907 |
| 8 | Polynomial | 0.9283 | 12.0330 | 14.9227 | 0.9281 | 13.7164 | 16.4672 |
| | **PUK** | **0.9398** | **9.7984** | **13.0830** | **0.9392** | **11.2408** | **15.5611** |
| | Radial basis function | 0.9283 | 18.6912 | 22.9160 | 0.9281 | 19.1426 | 25.6111 |
| 9 | Polynomial | 0.8394 | 17.2037 | 21.1520 | 0.8380 | 17.9619 | 22.8538 |
| | PUK | 0.8755 | 14.3397 | 18.8555 | 0.8623 | 16.2552 | 20.9296 |
| | Radial basis function | 0.8398 | 25.6982 | 31.0532 | 0.8380 | 26.4915 | 31.2574 |
| 10 | Polynomial | 0.8777 | 14.7758 | 19.8128 | 0.8746 | 15.2039 | 19.8289 |
| | PUK | 0.9087 | 12.2525 | 17.3738 | 0.9084 | 12.0109 | 16.8281 |
| | Radial basis function | 0.8779 | 23.2745 | 28.4086 | 0.8748 | 23.7460 | 28.7051 |

The scatter plot and time series graph drawn for the SVR model are given in Figures 6 and 7, which shows that all points are compatible with FAO56PM - $ET_0$ values and $ET_0$ values estimated from the model, except for the less frequent endpoints. The $R^2$ values were also very high ($R^2 > 0.939$).

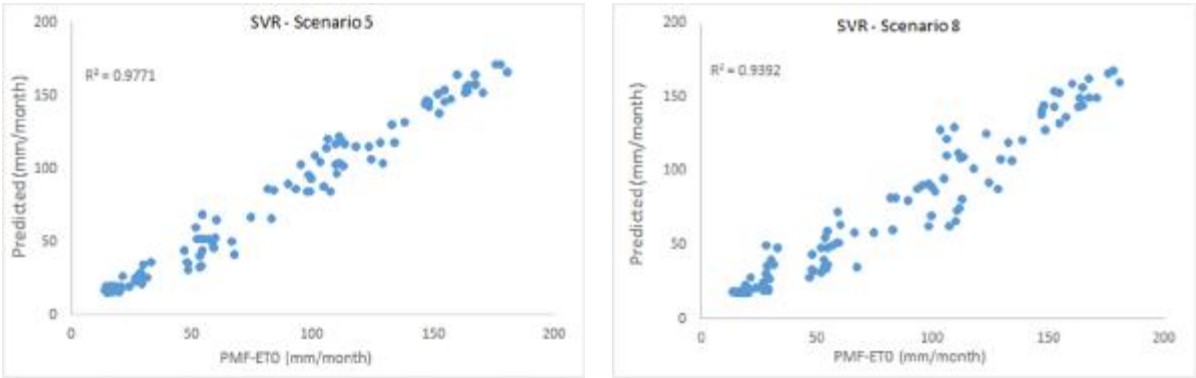

Figure 6. Scatter plots comparing SVR estimated and FAO56PM estimated $ET_0$ in scenarios 5 and 8

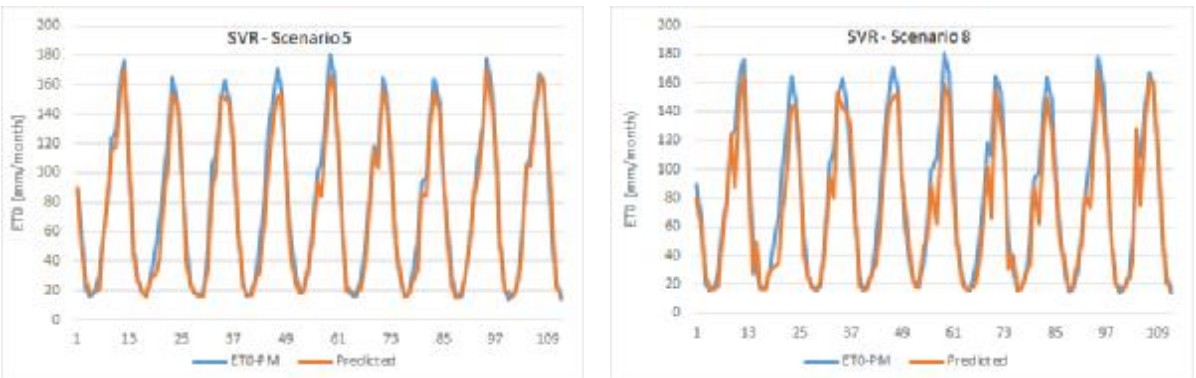

Figure 7. Time series graphics of SVR estimated and FAO56PM estimated $ET_0$ in scenarios 5 and 8

In this study, the BFGS training algorithm was specifically used to train the ANN architecture and $ET_0$ amounts were estimated for all scenarios. The results are given in Table 6. In implementing the BFGS-ANN method, all features, including the target feature (meteorological variables and $ET_0$) are standardized. In the hidden and output layer, the sigmoid function is $f(x) = 1 / (1 + e^{-x})$ used for classification.

As can be seen here, scenarios 5 and 8 gave the best and most relevant results. According to the results, the 5th scenario including TMean, TMin, TMax and $n$ meteorological variables again produced the best result (Train period: $R^2 = 0.9843$, MAE = 8.0025 mm/month, RMSE = 9.9407 mm/month; Test period: $R^2 = 0.9781$, MAE = 6.7885 mm/month RMSE = 8.8991 mm/month). However, Scenario 8 gave the most appropriate result (Train period: $R^2 = 0.9474$, MAE = 10.1139 mm/month, RMSE = 13.1608 mm/month; Test period: $R^2 = 0.9428$, MAE = 11.4761 mm/month, RMSE = 15.6399 mm/month) with only the sunshine duration ($n$) meteorological input variable, hence been the optimally selected BFGS-ANN model. Although the

8th scenario's accuracy rate is marginally less than the 5th scenario, it is easy and practical in terms of application and calculation since it consists of only one parameter. The scatter plot and time series graph drawn for the BFGS-ANN model,

given in Figures 8 and 9 concurs with the statistical metrics of Table 6. As can be seen, the BFGS-ANN method predicted $ET_0$ amounts with a high success rate, and a high level of agreement was achieved between the estimates obtained from the model and FAO56PM- $ET_0$ values. The $R^2$ values were also very high ($R^2 > 0.942$).

Table 6. Outcomes of the BFGS-ANN modelling approach for different Scenarios based on $R^2$, MAE, and RMSE (*Italics*

represents the best results; Bold represents the optimally selected model).

| Scenario No | Train | | | Test | | |
|---|---|---|---|---|---|---|
| | $R^2$ | MAE | RMSE | $R^2$ | MAE | RMSE |
| 1 | 0.9778 | 6.7017 | 8.6972 | 0.9769 | **6.6346** | **8.6243** |
| 2 | 0.9763 | 7.2683 | 9.6751 | 0.9700 | 7.5305 | 10.3722 |
| 3 | 0.9450 | 9.2810 | 12.3463 | 0.9423 | 11.2870 | 14.3732 |
| 4 | 0.9670 | 7.8325 | 10.4035 | 0.9659 | 9.1159 | 12.4740 |
| 5 | *0.9843* | 8.0025 | *9.9407* | *0.9781* | *6.7885* | *8.8991* |
| 6 | 0.9536 | 8.9027 | 11.3546 | 0.9522 | 11.5089 | 14.7687 |
| 7 | 0.9466 | 10.2246 | 13.2535 | 0.9417 | 11.9444 | 15.7787 |
| 8 | **0.9474** | **10.1139** | **13.1608** | **0.9428** | **11.4761** | **15.6399** |
| 9 | 0.8768 | 14.8765 | 18.4766 | 0.8709 | 15.9139 | 19.8957 |
| 10 | 0.9158 | 13.0161 | 16.2424 | 0.9149 | 12.4874 | 15.5428 |

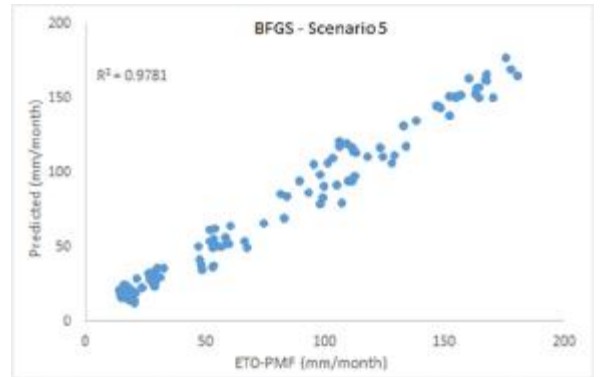 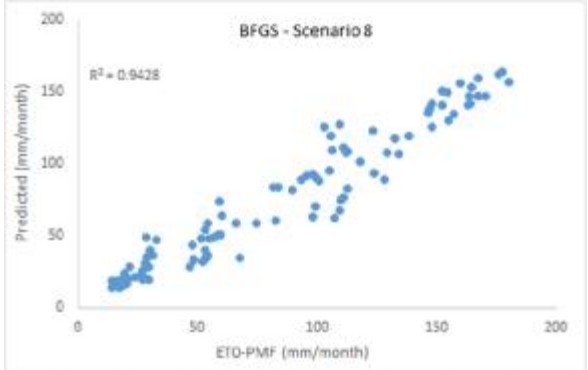

Figure 8. Scatter plots comparing BFGS-ANN estimated and FAO56PM estimated $ET_0$ in scenarios 5 and 8

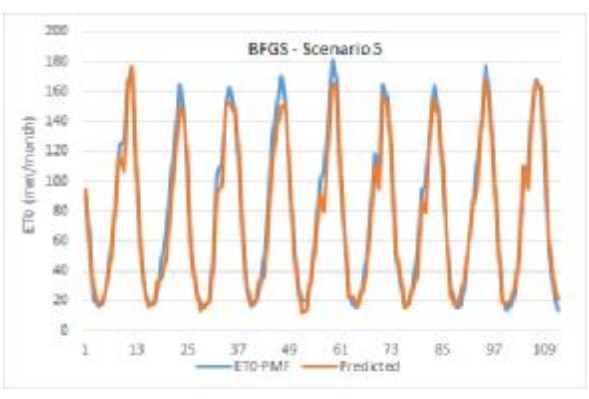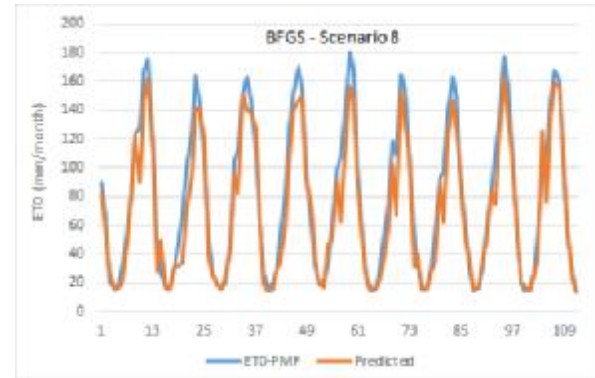

Figure 9. Time series graphics of BFGS-ANN estimated and FAO56PM estimated $ET_0$ in scenarios 5 and 8

Finally, the LSTM method, which is a deep learning technique, was used to estimate the $ET_0$ under the same 10 scenarios. Two hidden layers with 200 and 150 neurons were utilized in LSTM with the rectified linear unit (ReLU) activation function and Adam optimizations. The other parameters: Learning rate alternatives from $1e^{-1}$ to $1e^{-9}$, Decay as $1e^{-1}$ to $1e^{-9}$, and 500-750-1000 as epochs have been tried. The best results obtained for 10 different scenarios at the modelling stage, according to the LSTM method, are given in Table 7.

Table 7. Outcomes of the LSTM modelling approach for different Scenarios based on $R^2$, MAE, and RMSE (*Italics* represents the best results; Bold represents the optimally selected model).

| Scenario No | Train | | | Test | | |
|---|---|---|---|---|---|---|
| | $R^2$ | MAE | RMSE | $R^2$ | MAE | RMSE |
| 1 | 0.9825 | 7.0178 | 9.3020 | 0.9769 | 8.6232 | 11.4663 |
| 2 | 0.9618 | 9.0678 | 12.4321 | 0.9604 | 8.5703 | 11.7467 |
| 3 | 0.9403 | 13.841 | 16.3260 | 0.9345 | 14.8644 | 17.1128 |
| 4 | 0.9499 | 10.375 | 12.3748 | 0.9393 | 11.5043 | 13.7417 |
| 5 | *0.9835* | *4.9405* | *6.8687* | *0.9759* | *6.2907* | *8.5897* |
| 6 | 0.9694 | 11.532 | 15.7447 | 0.9602 | 8.1580 | 10.6059 |
| 7 | 0.9382 | 10.962 | 14.8716 | 0.9366 | 10.1113 | 13.6070 |
| 8 | **0.9461** | **12.461** | **15.7539** | **0.9384** | **11.6711** | **14.4864** |
| 9 | 0.8807 | 14.479 | 18.2882 | 0.8664 | 15.2565 | 19.4120 |
| 10 | 0.9231 | 14.195 | 17.1729 | 0.9220 | 13.7034 | 16.1857 |

As in other methods, the 5th and 8th scenarios of the LSTM model registered the best and most appropriate results. In the 5th scenario TMean, TMin, TMax and *n* as the input variables gave the best result (Train period: $R^2$ = 0.9835, MAE = 4.9405

mm/month, RMSE = 6.8687 mm/month; Test period: $R^2$ = 0.9759, MAE = 6.2907 mm/month RMSE = 8.5897 mm/month).
However, scenario 8 gave the most appropriate result (Train period: $R^2$ = 0.9461, MAE = 12.461 mm/month, RMSE = 15.7539 mm/month; Test period: $R^2$ = 0.9384, MAE = 11.6711 mm/month, RMSE = 14.4864 mm/month) with the sunshine duration (*n*) meteorological variable as the input to the model.

Scatter plot and time-series graphs of observed and LSTM predicted $ET_0$ are given in Figures 10 and 11, where again a high success rate, and a high level of agreement was achieved between the estimates obtained from the model and FAO56PM- $ET_0$ values.

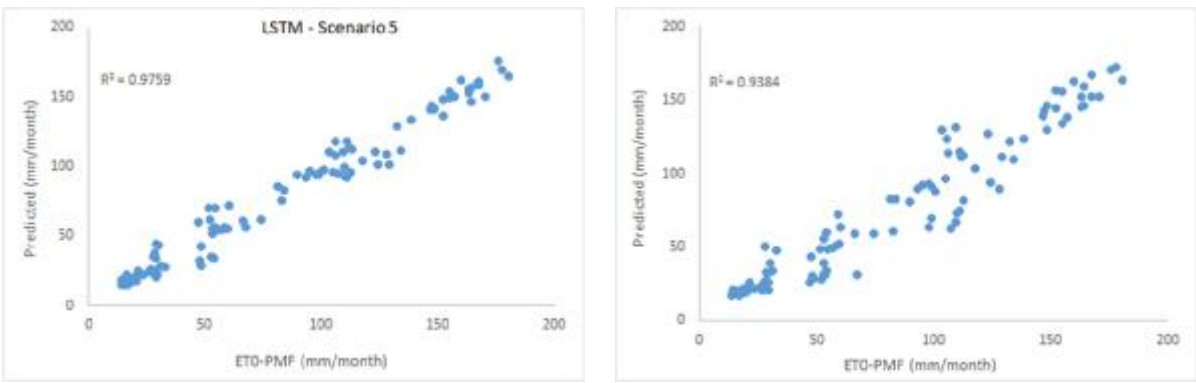

Figure 10. Scatter plots comparing LSTM estimated and FAO56PM estimated $ET_0$ in scenarios 5 and 8

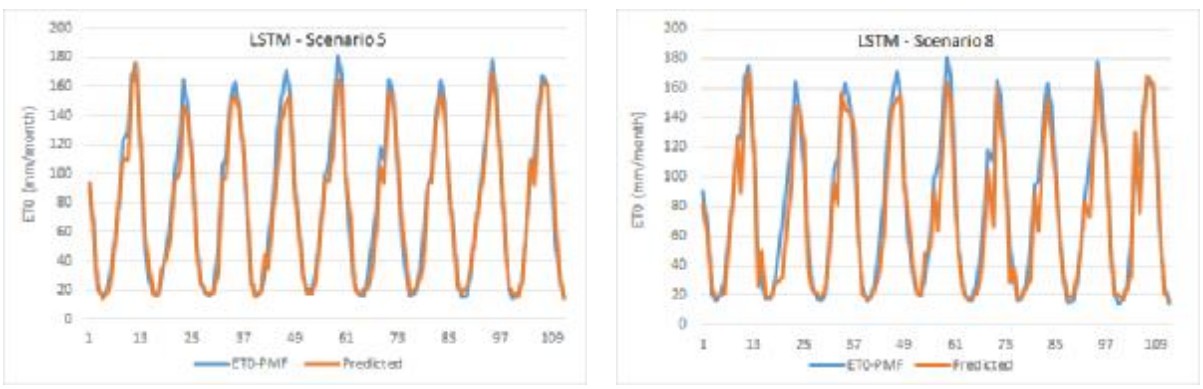

Figure 11. Time series graphics of LSTM estimated and FAO56PM estimated $ET_0$ in scenarios 5 and 8

In order to compare and evaluate the models used in this study, statistical values for the test phase are given in both FAO56PM-$ET_0$ and from the respective models in Table 8. The lowest skewness coefficient was found in scenario 5 in both GPR and SVR methods with 0.39 and the highest in LSTM scenario 8 with 0.52. The lowest kurtosis coefficient has Tmean with -1.23 and the highest with 0.36 by RHmean parameter. The highest variation was observed in RHmin with 174.19 and the lowest in U parameter with 0.17.

Table 8. Statistical values of the test phase for selected scenarios (Bold ones are the best/closest results).

| Statistic | GPR | | SVR | | BFGS-ANN | | LSTM | | ET$_0$PM |
| --- | --- | --- | --- | --- | --- | --- | --- | --- | --- |
| | Scenario 5 | Scenario 8 | Scenario 5 | Scenario 8 | Scenario 5 | Scenario 8 | Scenario 5 | Scenario 8 | |
| Minimum | 17.687 | 19.1090 | 15.1900 | 17.1520 | 12.2480 | **13.9060** | 14.2971 | 16.9787 | 13.99 |
| Maximum | 163.440 | 158.557 | **180.530** | 167.527 | 176.765 | 164.100 | 175.613 | 172.767 | 180.53 |
| Mean | **75.8818** | 71.3861 | 74.5771 | 71.2124 | 75.8644 | 70.7299 | 75.6023 | 72.3210 | 79.21 |
| Stdev | 48.8941 | 47.6359 | 51.5342 | 48.9192 | 50.6812 | 48.2539 | 50.0143 | **50.2075** | 53.26 |
| Correlation | 0.9820 | 0.9691 | 0.9885 | 0.9691 | **0.9890** | 0.9710 | 0.9879 | 0.9687 | 1 |
| Skewness | 0.39 | 0.47 | 0.39 | 0.51 | 0.41 | 0.46 | **0.36** | 0.52 | 0.36 |
| Kurtosis | -1.29 | **-1.27** | -1.32 | -1.16 | -1.24 | -1.21 | -1.21 | -1.16 | -1.27 |
| Coefficient of variation | 2344.09 | 2226.93 | **2655.77** | 2393.09 | 2568.59 | 2328.44 | 2501.43 | 2520.80 | 2836.65 |
| Number of records | 112 | 112 | 112 | 112 | 112 | 112 | 112 | 112 | 112 |

As can be seen from Table 8, the closest value to the FAO56PM-ET$_0$ minimum value (13.99 mm/month) is the 8th scenario in the BFGS-ANN method (13.906 mm/month). Furthermore, the FAO56PM-ET$_0$ maximum value (180.53 mm/month) has been reached in the 5th scenario (180.53 mm/month) in the SVR method which is the closest and even the same value. The value closest to the mean value of FAO56PM-ET$_0$ (79.21 mm/month) corresponds to the 5th scenario (75.8818 mm/month) in the
420 GPR method; the value closest to the FAO56PM-ET$_0$ Stdev value (53.26 mm/month) is the value of the 5th scenario (51.5342 mm/month) in the SVR method. As shown in Table 8, all methods have estimated the ET$_0$ amounts within acceptable, yet disparate results are attained when comparing the statistics. Having said that, when models are ranked according to the correlation coefficient, the best results were BFGS-ANN, SVR, LSTM, and GPR in the 5th scenario and BFGS-ANN, GPR, SVR, and LSTM in the 8th scenario.
Furthermore, to have precise model comparative evaluations besides the tables, the Taylor diagram for the 5th and 8th scenarios were plotted as in Figure 12. The points on the polar Taylor graph are used to study the adaption between measured and predicted values in the Taylor diagram. The correlation coefficient and normalized standard deviation are also indicated by the azimuth angle, and radial distances from the base point, respectively (Taylor, 2001). As displayed in the figure, all four models performed quite well but BFGS-ANN seemed to achieve higher success than others. As stated earlier in Figure 1- histogram,
it is seen that FAO56PM- ET$_0$ values do not conform to normal distribution. This mismatch is considered to be the reason for the poor performances of the GPR method over comparative models.

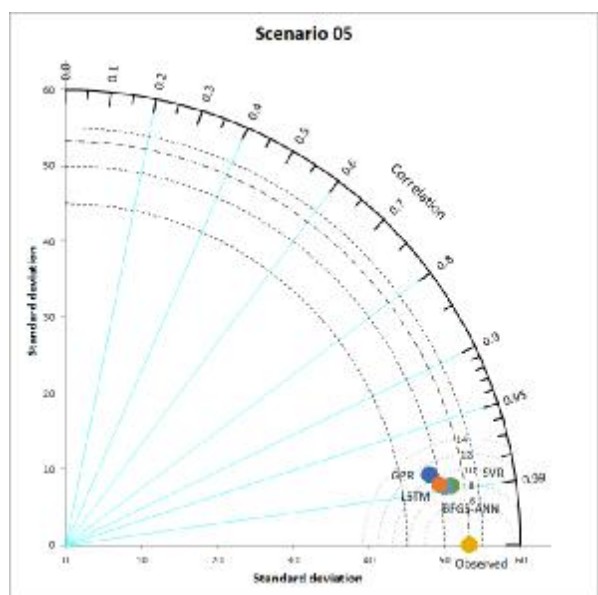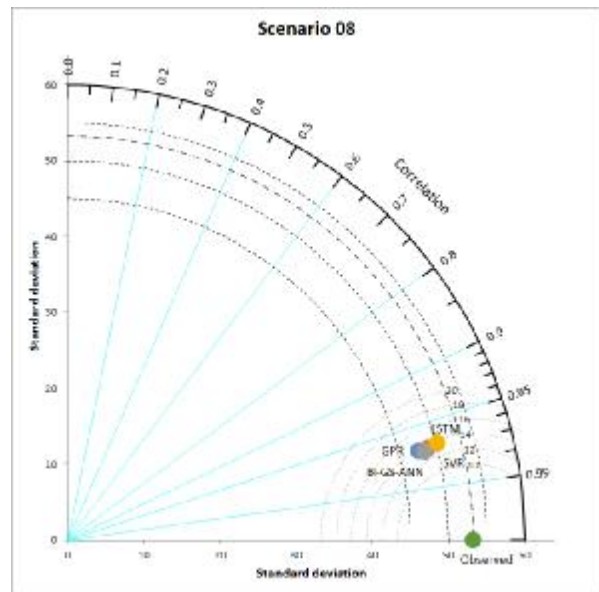

Figure 12. Taylor diagrams of scenarios 5 and 8

The results of Figure 12 also show that models performances were higher in Scenario 5, however, using the least input parameters to develop the most parsimonious model was the key target of the study and was achieved by Scenario 8 whereby ET$_0$ values were estimated correctly at relatively appropriate and acceptable levels. Therefore, these methods produced trustworthy results and have the potential to make correct estimations in climates similar to the study area.

**5 Conclusion**

The amount of ET$_0$ can be calculated with many empirical equations. However, these equations can generally differ spatially and require the knowledge of many parameters. Since ET$_0$ includes a complex and nonlinear structure, it cannot be easily estimated with the previously measured data without requiring numerous parameters. In this study, estimating the ET$_0$ with different machine learning and deep learning methods was made using the least meteorological variable in Turkey's Corum region, with an arid and semi-arid climate regarded as a strategic agricultural region. In this context, firstly, ET$_0$ amounts were calculated with the Penman-Monteith method and taken as the output of the models. Then, 10 different scenarios were created using different combinations of meteorological variables. Consequently, Kernel-based GPR and SVR methods, BFGS-ANN, and LSTM models were developed for monthly ET$_0$ amount estimations. The results revealed better performance of the BFGS-ANN model in comparison to other models under this study, although all four methods predicted ET$_0$ amounts within acceptable accuracy and error levels. In kernel-based methods (GPR and SVR), PUK was the most successful kernel function. The 5[th] scenario, which is related to temperature and includes four meteorological variables (mean temperature, highest and

lowest temperature averages, and sunshine duration) gave the best results in all the scenarios used. Scenario 8, which included
only the sunshine duration, was determined as the most suitable and parsimonious scenario. In this case, the $ET_0$ amount was estimated using only sunshine duration without the need for other meteorological parameters for the study area. The Corum region is described as arid and semi-arid with low rainfalls and cloudiness and longer sunshine durations, hence sunshine hours is the key driving factor of $ET_0$ in the region which is clearly highlighted by high model performances with sunshine hours as the only input. Continuous measurement of meteorological variables in large farmland is a costly process that requires expert
personnel, time, or good equipment. Simultaneously, some equations used for $ET_0$ calculations are not preferred by specialists because they contain many parameters. In this case, it is very advantageous for water resources managers to estimate $ET_0$ amounts only with sunshine duration time, which is easy to measure and requires no extra cost. The follow-up study aims to evaluate the performance of GPR and LSTM models in a larger area on a daily time scale and with data to be obtained from more meteorology stations.

**Funding:**

*This work was supported by Fellowships for Visiting Scientists and Scientists on the Sabbatical Leave Programme (2221) of The Scientific and Technological Research Council of Turkey (TUBITAK).*

**Acknowledgement:**
Authors acknowledge the 'Open Access Funding by the Publication Fund of the TU Dresden'.

**Author Contributions: Conceptualization, MTS, and HA; Data curation, MTS, HA; Formal analysis, SS; Funding acquisition, HA, MTS; Investigation, MTS, HA and SS; Methodology, MTS, HA and SS; Project administration, HA; Resources, MTS and HA; Software, MTS, AM and HA; Supervision, MTS, HA.; Validation, AM and SS; Visualization, MTS, HA, SS and RP;**
**Writing—original draft, MTS, HA and SS; Writing—review and editing, RP, MTS, AM and SS. All authors have read and agreed to the published version of the manuscript.**

**Conflicts of Interest:** The authors declare no conflict of interest.

**Code/Data availability**: Data are available on request due to privacy or other restrictions.

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
