# Peer review of "Comparative analysis of Kernel-based versus BFGS-ANN and deep learning methods in monthly reference evapotranspiration estimation"

_Hydrology and Earth System Sciences, 2020_

## Short Comment (SC1) · 11 Aug 2020

1- Share more numerical results in the Abstract. Mention the numerical results of the Model Evaluation in the Abstract. 2- Table 1 description is not enough. Add the skewness coefficient, kurtosis coefficient and the coefficient of variation in Table 1. 3- Show the study area on the Map of Turkey. 4- The figures on figure 1 are not read. Please put in legible shape. 5- Add the coefficients of determination to the graphs in Figure 2. 6- Substitute the results of the determination coefficient for the correlation coefficient, in table 4, 5, 6, and 7. 7- Add the coefficients of determination

to the graphs in Figure 4-6-8-10. 8- The resolution of all the figures in the study is low, the figures are not readable. 9- Add the skewness coefficient, kurtosis coefficient and the coefficient of variation in Table8. 10- Important and relevant references have been ignored: Cobaner, M., CitakoÄ§lu, H., Haktanir, T., & Kisi, O. (2017). Modifying Hargreaves–Samani equation with meteorological variables for estimation of reference evapotranspiration in Turkey. Hydrology Research, 48(2), 480-497.

---

## Referee Comment (RC1) · Anonymous Referee #1 · 17 Aug 2020

1- Share more numerical results in the Abstract. Mention the numerical results of the Model Evaluation in the Abstract. 2- Table 1 description is not enough. Add the skewness coefficient, kurtosis coefficient and the coefficient of variation in Table 1. 3- Show the study area on the Map of Turkey. 4- The figures on figure 1 are not read. Please put in legible shape. 5- Add the coefficents of determination to the graphs in Figure 2. 6- Substitute the results of the determination coefficient for the correlation coefficient, in table 4, 5, 6, and 7. 7- Add the coefficents of determination to the graphs in Figure 4-6-8-10. 8- The resolution of all the figures in the study is

low, the figures are not readable. 9- Add the skewness coefficient, kurtosis coefficient and the coefficient of variation in Table 8. 10- Important and relevant references have been ignored: Cobaner, M., Citakoglu, H., Haktanir, T., & Kisi, O. (2017). Modifying Hargreaves–Samani equation with meteorological variables for estimation of reference evapotranspiration in Turkey. Hydrology Research, 48(2), 480-497.

---

## Referee Comment (RC2) · Anonymous Referee #2 · 24 Sep 2020

The study estimated monthly reference evapotranspiration (ET0) using four different machine learning techniques, including Gaussian process regression (GPR), support vector regression (SVR), long short-term memory (LSTM), and artificial neural network with the training function of Broyden-Fletcher-Goldfarb-Shanno quasi-Newton (BFGS-ANN). To obtain the best modeling performance, three different kernel functions for both GPR and SVM, and ten different combinations as inputs for all the models proposed were evaluated, respectively. LSTM method is currently an extensively used method in literature to address nonlinear regression problems in a wide range of applications. LSTM was compared with three conventional approaches (ANN, SVM and GPR), which provides a good and new insight to the existing studies. Regrettably, these models were not well investigated in terms of their generalization ability and computational efficiency. Moreover, the manuscript was not well-written, and its shortcomings can be found in each section. Substantial language improvements should be also made. Therefore, the manuscript needs major revisions before I can recommend it for publication.

Major Comments: 1). Sections Introduction and Methods were not well-written, as well as the organization and design of figures and tables. 2). The dataset was split into two parts for training and testing. However, the results of all figures and tables were only shown in the testing period. 3). As we all know, machine learning is being widely used for addressing many issues, mainly including classification and regression. This study was conducted for regression and aimed at modeling and predicting monthly ET0. I don't know why the descriptions related to classification and classifier were frequently shown.

Introduction 1). For the first paragraph, is it a popular science article? Or suggest deleting this paragraph. 2). Lines 41-52: Some classical previous studies and reviews should be cited for support these descriptions. Besides, it is well known that many physical and empirical models as common methods have been widely used to estimate ET0. Suggest pointing out their advantages and disadvantages, and give some reasons why artificial intelligence (AI) techniques were adopted as alternative tools for this work. 3). As shown in Lines 53-119, so many previous studies (18) of ET0 estimation using different artificial intelligence models were reviewed monotonously. It is utterly pointless. Why did you carry out this study? It should be supported by more sound reasons. Suggest focusing on reviewing some extensively methods (e.g., ANN, SVM, GRNN) for ET0 prediction, and point out their advantages and disadvantages when estimating ET0 in terms of their performance and computational efficiency. For example, both ANN and SVM methods have received a great deal of attention in the

last decade and have been extensively utilized in diverse fields. Nevertheless, these two approaches still have some shortcomings, which have been revealed by previous studies. In general, the ability of ANN method is limited by several disadvantages, such as slow learning speed, over-fitting and local minima. Additionally, it is also relatively difficult to determine some key parameters, such as training function and activation function. SVM also exists several drawbacks, such as high memory requirement and a large amount of computing time during learning process. In order to overcome the disadvantages of these two approaches, many new modeling techniques have been proposed in recent years. For instance, two state-of-the-art machine learning techniques, namely LSTM and GPR, are widely utilized in the hydrologic time series modeling and forecasting. To the best of our knowledge, however, there have been very few attempts to test the practicability and ability of these two advanced approaches (LSTM and GPR) for ET0 modeling and prediction. 4). Regarding the last paragraph, the comparison of different kernel functions for SVM and GPR models, was designed as one goal of this study. Why did you attempt to compare these kernel functions? This aim should be supported by more sound reasons. To the best of my knowledge, many similar studies have been reported, which should reviewed before this paragraph. 5). For ANN model, training function plays an important role in its generalization performance. To my knowledge, a number of training functions (>10) can be used as alternative inner functions, such as conjugate gradient algorithms, gradient descent methods, quasi-Newton methods, Bayesian regulation backpropagation and one step secant backpropagation. The effects of these training functions on ANN have been reported frequently in in diverse fields. These related studies should be reviewed for offering more sound reasons for this paper. More importantly, in this study, why was BFGS selected as training function for ANN model? In order to better check the performance of these training functions, more training functions also can be adopted and compared with BFGS algorithm in this work. Materials and methods 1). Check the titles of "2 Material and method" and "3 Methods". 2). For Table1, to better compare and evaluate the performance of the used models, the statistics of the data should be divided according to training and testing

periods.

Methods 1). For each method used in this paper, many irrelevant descriptions and inessential details should be omitted. More rigorous and precise description about the principle of the method used in this study should be given. Furthermore, some important and classical papers should be cited. 2). For each method, please point out some special inner functions and parameters of the developed models. Because different functions and parameters have great effects on the generalization of those models. Taking ANN method for example, its generalization performance is generally dependent on many factors, mainly including topological structure of network and relevant parameters (e.g., learning rate, regularization factor and momentum factor) and functions (e.g., learning, activation and training algorithms). In this study, apart from training algorithm, the remaining features above-mentioned were determined by the trial and error method. 3). Suggest adding some descriptions about the used toolbox, package or software for each method.

Results 1). The descriptions of all the tables and figures were so simple and monotonous. 2). As the title of this section is shown, more discussion should be given about this study.

Conclusion 1). In this study, ET0 and its related meteorological data at a time scale of month were gathered from one weather station. Results showed that all the proposed models did a good job in simulating monthly ET0. Nevertheless, these machine learning methods are likely to be questioned in that the intrinsic mechanisms of these well-trained black box models remain poorly described or understood. To a certain degree, this limitation decreases the reliability of these techniques. 2). In the follow-up work, the performance of the GPR and LSTM models for the present study should be further evaluated at finer time scales, such as daily. Moreover, more weather stations or regions should be taken into consideration.

224, 2020.

---

## Author Comment (AC1) · 26 Sep 2020

Dear Citakoglu, Thank you for your valuable and kind comments. We agree with all your comments. 1) We will give more numerical results in the summary section. 2) Related coefficients (the skewness, kurtosis and variation) will be added to Table 1. 3) A map showing the study area will be added. 4) The quality of Figure 1 will be improved. 5) Coefficients of determination will be added to Figure 2. 6) Necessary changes will be made in Table 4-7. 7) Coefficients of determination will be added to Figure 4-6-8-10. 8) The resolution of figures will be improved. 9) Related coefficients

(the skewness, kurtosis and variation) will be added to Table 8. 10) Relevant reference will be considered.

We are thankful to Ms. Citakoglu for her valuable comments towards the improvement of our paper.

---

## Author Comment (AC2) · 26 Sep 2020

Dear Referee, Thank you for your valuable and kind comments. We agree with all your comments. 1) We will give more numerical results in the summary section. 2) Related coefficients (the skewness, kurtosis and variation) will be added to Table 1. 3) A map showing the study area will be added. 4) The quality of Figure 1 will be improved. 5) Coefficients of determination will be added to Figure 2. 6) Necessary changes will be made in Table 4-7. 7) Coefficients of determination will be added to Figure 4-6-8-10. 8) The resolution of figures will be improved. 9) Related coefficients

(the skewness, kurtosis and variation) will be added to Table 8. 10) Relevant reference will be considered.

We are thankful to the Referee for her valuable comments towards the improvement of our paper.

---

## Author Comment (AC3) · 7 Nov 2020

Dear Editor/Reviewer,

PS. Our responses to the referee suggestions are also provided as supplement to make them visual.

Dear Referee, Thank you for your valuable and kind comments. We agree with all your comments. 1- Share more numerical results in the Abstract. Mention the numerical results of the Model Evaluation in the Abstract. Response: More numerical results have

been given in the summary. 2- Table 1 description is not enough. Add the skewness coefficient, kurtosis coefficient and the coefficient of variation in Table 1. Response: More comments have been added related Table 1. The skewness, kurtosis and variation coefficients have been added to Table 1. 3- Show the study area on the Map of Turkey. Response: A map (Figure 1) showing the study area has been added. 4- The figures on figure 1 are not read. Please put in legible shape. Response: The quality of Figure 1 (Figure 2) has been improved. 5- Add the coefficients of determination to the graphs in Figure 2. Response: Coefficients of determination have been added to Figure 2 (Figure 3). 6- Substitute the results of the determination coefficient for the correlation coefficient, in table 4, 5, 6, and 7. Response: The correlation coefficients in Table 4-7 have been replaced by the determination coefficient. Accordingly, necessary corrections have been made in the text. 7- Add the coefficients of determination to the graphs in Figure 4-6-8-10. Response: Coefficients of determination have been added to Figure 4-6-8-10. 8- The resolution of all the figures in the study is low, the figures are not readable. Response: The resolution of figures have been improved. 9- Add the skewness coefficient, kurtosis coefficient and the coefficient of variation in Table 8. Response: The skewness, kurtosis and variation coefficients have been added to Table 8. 10- Important and relevant references have been ignored: Cobaner, M., Citakoglu, H., Haktanir, T., & Kisi, O. (2017). Modifying Hargreaves–Samani equation with meteorological variables for estimation of reference evapotranspiration in Turkey. Hydrology Research, 48(2), 480-497.

Response: Relevant reference has been examined and added to the article as a reference.

We are thankful to the Referee for her valuable comments towards the improvement of our paper.

Please also note the supplement to this comment:
https://hess.copernicus.org/preprints/hess-2020-224/hess-2020-224-AC3-

supplement.pdf

---

## Editor Comment (EC1) · Dimitri Solomatine (Editor) · 19 Nov 2020

Quite a number of comments, and very useful ones. The authors have In their answers addressed most of them (albeit not always convincingly in my view), but obviously there is still a lot to do in revising the manuscript. Clearly the authors have to convincingly show why ML methods are to be used, along with the traditional methods. PLease address every comment, and if you disagree, please provide justified convincing rebuttal, arguments why. Please show in the manuscript how have you responded to comments by highlighting the changes. Please ensure that it isnot only referees who

have to be answered but also the readers who need to receive the answers to the questions/comments posed/made by referees. Success in preparing the revised version!

---

## Author Response (AR1)

**Reviewer 1**

Response:

Dear Referee,

Thank you for your valuable and kind comments. We agree with all your comments.

1- Share more numerical results in the Abstract. Mention the numerical results of the Model Evaluation in the Abstract.

Response: More numerical results have been given in the Abstract.

2- Table 1 description is not enough. Add the skewness coefficient, kurtosis coefficient and the coefficient of variation in Table 1.

Response: The skewness, kurtosis and variation coefficients have been added in Table 1 for training data set. Additionally, these parameters for testing data have also been included for brevity. The discussions pertaining to the statistics has also been included in the text.
Please see lines 170-180.

3- Show the study area on the Map of Turkey.

Response: A map showing the study area has been added as Figure 1. Relevant discussions have been added in the text in Section 2: Study area and dataset used

4- The figures on figure 1 are not read. Please put in legible shape.

Response: The quality of Figure 1 (Figure 2) has been improved.

5- Add the coefficients of determination to the graphs in Figure 2.

Response: Coefficients of determination have been added to Figure 2 (Figure 3) and the discussion have been provided in the main text.

6- Substitute the results of the determination coefficient for the correlation coefficient, in table 4, 5, 6, and 7.

Response: The correlation coefficients in Table 4-7 have been replaced by the determination coefficient. Accordingly, necessary corrections have been made in the text.

7- Add the coefficients of determination to the graphs in Figure 4-6-8-10.

Response: Coefficients of determination have been added to Figure 4-6-8-10.

8- The resolution of all the figures in the study is low, the figures are not readable.

Response: The resolution of figures have been improved.

9- Add the skewness coefficient, kurtosis coefficient and the coefficient of variation in Table 8.

Response: The skewness, kurtosis and variation coefficients have been added to Table 8 with relevant discussions in lines 450-455

10- Important and relevant references have been ignored: Cobaner, M., Citakoglu, H., Haktanir, T., & Kisi, O. (2017). Modifying Hargreaves–Samani equation with meteorological variables for estimation of reference evapotranspiration in Turkey. Hydrology Research, 48(2), 480-497.

Response: Relevant reference has been examined and added to the article as a reference.

We are thankful to the Referee for her valuable comments towards the improvement of our paper.

**Reviewer 2**

The study estimated monthly reference evapotranspiration (ET0) using four different machine learning techniques, including Gaussian process regression (GPR), support vector regression (SVR), long short-term memory (LSTM), and artificial neural network with the training function of Broyden-Fletcher-Goldfarb-Shanno quasi-Newton (BFGSANN). To obtain the best modeling performance, three different kernel functions for both GPR and SVM, and ten different combinations as inputs for all the models proposed were evaluated, respectively. LSTM method is currently an extensively used method in literature to address nonlinear regression problems in a wide range of applications. LSTM was compared with three conventional approaches (ANN, SVM and GPR), which provides a good and new insight to the existing studies. Regrettably, these models were not well investigated in terms of their generalization ability and computational efficiency. Moreover, the manuscript was not well-written, and its shortcomings can be found in each section. Substantial language improvements should be also made. Therefore, the manuscript needs major revisions before I can recommend it for publication.

Response:

Dear Referee,

Thank you for your valuable and kind comments. In this study, we will take your valuable comments into account as much as we can. We hope that your feedback will help us improve the quality of the study.

**Major Comments:**
1). Sections Introduction and Methods were not well-written, as well as the organization and design of figures and tables.
Response: The paper has received in-depth language review from a native-speaker PhD in Applied Linguistics, in full command of academic English. We have re-done the whole introduction making it succinct and revealing the research gaps, while the whole paper has been improved and proof read properly.

2). The dataset was split into two parts for training and testing. However, the results of all figures and tables were only shown in the testing period.
Response: In artificial intelligence-based studies, the evaluation of model performances is based on the performance of the model during the testing period. The purpose of doing this is to assess the models' generalization ability on unseen (test) data set. This is the desired methodology and has been adopted in this study. Nonetheless, since very successful results were obtained in our study, at the referee's recommendation, the evaluation metrics for the train period have been added to the table 4-5-6-7 as well.

3). As we all know, machine learning is being widely used for addressing many issues, mainly including classification and regression. This study was conducted for regression and aimed at modeling and predicting monthly ET0. I don't know why the descriptions related to classification and classifier were frequently shown.
Response: In this study regression was, of course, what was mainly conducted. Largely, Weka software was used for regression and predictions. Descriptions of classification and classifier are an important aspect of machine learning based modelling approaches in order to have better clarity of the data, conditions used, outcomes to avoid spurious relations and non-credible results. This is an acceptable practice used in all machine learning based applications. The outcomes then can easily be replicated, if the decision makers are interested in adopting any of the models.

**Introduction**
1). For the first paragraph, is it a popular science article? Or suggest deleting this paragraph.
Response: The paragraph as per the referee's recommendation has been deleted.

2). Lines 41-52: Some classical previous studies and reviews should be cited for support these descriptions. Besides, it is well known that many physical and empirical models as common methods have been widely used to estimate ET0. Suggest pointing out their advantages and disadvantages, and give some reasons why artificial intelligence (AI) techniques were adopted as alternative tools for this work.

Response: Some classical previous studies and reviews references have been added to (current) first three paragraphs. In the introduction, the advantages, disadvantages and the need for artificial intelligence in calculating ET0 have been emphasized more strongly.

3). As shown in Lines 53-119, so many previous studies (18) of ET0 estimation using different artificial intelligence models were reviewed monotonously. It is utterly pointless. Why did you carry out this study? It should be supported by more sound reasons. Suggest focusing on reviewing some extensively methods (e.g., ANN, SVM, GRNN) for ET0 prediction, and point out their advantages and disadvantages when estimating ET0 in terms of their performance and computational efficiency. For example, both ANN and SVM methods have received a great deal of attention in the last decade and have been extensively utilized in diverse fields. Nevertheless, these two approaches still have some shortcomings, which have been revealed by previous studies. In general, the ability of ANN method is limited by several disadvantages, such as slow learning speed, over-fitting and local minima. Additionally, it is also relatively difficult to determine some key parameters, such as training function and activation function. SVM also exists several drawbacks, such as high memory requirement and a large amount of computing time during learning process. In order to overcome the disadvantages of these two approaches, many new modeling techniques have been proposed in recent years. For instance, two state-of-the-art machine learning techniques, namely LSTM and GPR, are widely utilized in the hydrologic time series modeling and forecasting. To the best of our knowledge, however, there have been very few attempts to test the practicability and ability of these two advanced approaches (LSTM and GPR) for ET0 modeling and prediction.

Response: We agree with the referee's opinion on different artificial intelligence methods. Generally, ANN methods have disadvantages. However, despite all the disadvantages, it is still a preferred method in all branches of science and especially in Hydrology. In general, the capability of different ANN methods is discussed in terms of various disadvantages such as the large number of hidden layers, slow learning speed, overfitting and sticking to local minimums. At the same time, machine learning methods have some deficiencies such as the occasional difficulty of use, high memory requirements and large amounts of computation time in the learning process. In such a case, we see some recent developments in ANN methods and the use of deep learning techniques such as LSTM in water engineering. However, technical developments in computers and the emergence of relatively comfortable coding languages such as Python have enabled the overcoming of some deficiencies. In this study, we think that using different deep learning, machine learning and ANN methods in estimation of ET0 can shed light on future research and help determine more effective models in this field.
This opinion has been added at appropriate places in the introduction of the article.

4). Regarding the last paragraph, the comparison of different kernel functions for SVM and GPR models, was designed as one goal of this study. Why did you attempt to compare these kernel functions? This aim should be supported by more sound reasons. To the best of my knowledge, many similar studies have been reported, which should reviewed before this paragraph.

Response: Due to the nature of the data used in the study, it is seen that different models result in different accuracy rates. Changing the kernel function in machine learning methods (SVR and GPR) affects the results positively or negatively. In this study, we tried different functions in order to determine the best kernel function that is more compatible with the data we used and to increase the forecasting accuracy; hence the best function was determined using this method.

5). For ANN model, training function plays an important role in its generalization performance. To my knowledge, a number of training functions (>10) can be used as alternative inner functions, such as conjugate gradient algorithms, gradient descent methods, quasi-Newton methods, Bayesian regulation backpropagation and one step secant backpropagation. The effects of these training functions on ANN

have been reported frequently in in diverse fields. These related studies should be reviewed for offering more sound reasons for this paper. More importantly, in this study, why was BFGS selected as training function for ANN model? In order to better check the performance of these training functions, more training functions also can be adopted and compared with BFGS algorithm in this work.

Response: We certainly agree with the reviewer that there are many training functions that can possibly increase the performance of the ANN model and it is also possible to make a study comparing different training functions. However, our aim in this study is not to compare different training functions in ANN modeling. In the literature, ANN modeling using the BFGS function in ET0 prediction has been studied to a rather limited degree compared to other methods. In addition, the BFGS method is specified in the Weka document as a method that can yield quick results in cases where many parameters are involved. Citing these reasons, BFGS-ANN method has been included in this study. Instead of adding new training functions to the study in which GPR, SVR, ANN and LSTM methods were compared, we think that it would be more appropriate to compare only different training functions in a new study.

**Materials and methods**
1). Check the titles of "2 Material and method" and "3 Methods".
Response: Titles have been checked and corrected.

2). For Table1, to better compare and evaluate the performance of the used models, the statistics of the data should be divided according to training and testing periods.
Response: The statistics of the data have been divided according to training and testing periods in Table 1. Discussions on the training and test periods given in Table 1 have been added to the text.

**Methods**
1). For each method used in this paper, many irrelevant descriptions and inessential details should be omitted. More rigorous and precise description about the principle of the method used in this study should be given. Furthermore, some important and classical papers should be cited
Response: Less relevant definitions and details have been deleted.
Three important and classical papers (Chauhan and Shrivastava 2008, Kumar et al. 2002 and Allen 1989) and others have been cited in the in the introduction and methods section.

2). For each method, please point out some special inner functions and parameters of the developed models. Because different functions and parameters have great effects on the generalization of those models. Taking ANN method for example, its generalization performance is generally dependent on many factors, mainly including topological structure of network and relevant parameters (e.g., learning rate, regularization factor and momentum factor) and functions (e.g., learning, activation and training algorithms). In this study, apart from training algorithm, the remaining features above-mentioned were determined by the trial and error method.
Response: In this study, during SVR and GPR modeling, three kernel functions were used including Polynomial, Pearson VII function-based universal, and radial basis function with the level of Gaussian Noise Parameters added to the diagonal of the covariance matrix and the random number of seed to be used (equal to 1.0); the most suitable kernel function in each scenario was determined by trial and error.
BFGS-ANN method was also used for estimating ET0 values. This method was implemented on the basis of radial basis function networks trained in a fully supervised manner using WEKA's Optimization class by minimizing squared error with the BFGS method. In this method, all attributes are normalized into the [0,1] scale.
The initial centers for the Gaussian radial basis functions were found using WEKA's Simple K-Means. The initial sigma values were set to the maximum distance between any center and its nearest neighbour in the set of centers. Also in this method, one global sigma value was used for all units.
There were several parameters in the method. The ridge parameter was used to penalize the size of the weights in the output layer assumed to be 0.01, with the number of basis functions assumed to be two.

To improve speed, an approximate version of the logistic function was used as the activation function in the output layer.
All this description has been added to the article.

3). Suggest adding some descriptions about the used toolbox, package or software for each method.
Response: In this study, BFGS-ANN, SVR and GPR methods in the Weka software and Python language for LSTM method were used during modelling. This has been brought out in the method section of the text.

**Results**
1). The descriptions of all the tables and figures were so simple and monotonous.
Response: The titles and descriptions of the tables and figures have been checked and updated as much as possible.

2). As the title of this section is shown, more discussion should be given about this study.
Response: The interpretations were expanded by adding the results of the train period to the results of the existing test period.

**Conclusion**
1). In this study, ET0 and its related meteorological data at a time scale of month were gathered from one weather station. Results showed that all the proposed models did a good job in simulating monthly ET0. Nevertheless, these machine learning methods are likely to be questioned in that the intrinsic mechanisms of these well-trained black box models remain poorly described or understood. To a certain degree, this limitation decreases the reliability of these techniques.
Response: There are many references in the literature regarding the content and mathematical structure of the methods used in this study (Banda et al. 2018, Cobaner et al. 2017, Doorenbos and Pruitt 1977, Feng et al. 2017, Hargreaves and Samani 2013 etc.). Generally, these methods are used in many disciplines. The aim here is to evaluate the performance of different techniques in $ET_0$ estimations. One of the major shortcomings of these methods is that it is difficult to understand the complexity and even to use the methods. As a matter of fact, the changes that can be made in the structural content of the methods are very time-consuming and difficult, but can be scientifically beneficial.

2). In the follow-up work, the performance of the GPR and LSTM models for the present study should be further evaluated at finer time scales, such as daily. Moreover, more weather stations or regions should be taken into consideration.
Response: Following the referee's suggestion, in terms of recommendations for follow-up studies, it was stated in the conclusion section that the performance of the GPR and LSTM models stands to be evaluated in a larger area on a daily time scale and with data obtained from more meteorological stations.

We are thankful to the Referee for his/her valuable comments towards the improvement of our paper.

---

## Author Response (AR2)

Editor Decision: Publish subject to technical corrections (22 Dec 2020) by Dimitri Solomatine

Comments to the Author:
All major comments have been addressed, to the satisfaction of the referees.
I still suggest to go through the text, polishing English. Some of the corrections I would suggest to make (this is what I noticed, but there could be more):
L23: "machine and deep learning": deep learning is part of a wider area 'machine learning'... so I would just say 'machine learning'.
L20: drop "matter"
L 24: regardes -> is regarded
L439: "experimental" --> I think better to say "empirical"

Response:

Dear Solomatine,

Thank you for your valuable and kind comments. We agree with all your comments.

The paper was proofread properly and thoroughly to ensure that all grammatical errors and typos are removed wherever possible.

We are thankful to Dr. Solomatine for his valuable comments towards the improvement of our paper.

Editorial Office:

Please note that your reference list has not been compiled according to our standards. Please consider adjusting your reference list with the next revision of your manuscript. The manuscript preparation guidelines can be seen at: https://www.hydrology-and-earth-system-sciences.net/for_authors/manuscript_preparation.html.

Regarding figure 1: Please add the copyright symbol in the figure itself or in the caption as follows: © Google Maps.

I noticed that your tables contain coloured cells. Please note that this will not be possible in the final revised version of the paper due to HTML conversion of the paper. During the next file upload please update your manuscript accordingly. You can use footnotes or italic/bold font.

Response:

Dear Madam/Sir,

The reference list has been updated with respect to the manuscript preparation guidelines.

The copyright symbol has been added to figure 1

The colour highlights from the table has been removed and we have used italics and bold as suggested.

We are thankful to you for your valuable comments towards the improvement of our paper.